# Effectiveness of the Patient’s Severity Classification Competency Promotion Virtual Reality Program of Nursing Students during the COVID-19 Pandemic Period

**DOI:** 10.3390/healthcare11081122

**Published:** 2023-04-13

**Authors:** Eunju Lee, Gyuli Baek, Yeonhui Hwang

**Affiliations:** Nursing College, Keimyung University, Daegu 42601, Republic of Korea; 11578@kmu.ac.kr (E.L.); ja07079@naver.com (Y.H.)

**Keywords:** nursing students, patient severity classification competency, virtual reality, nursing education

## Abstract

The purpose of this study was to develop a virtual reality-based nursing education program aimed at improving nursing students’ severity classification competency. Severity classification in the emergency room is key to improving the efficiency of emergency room services worldwide. Prioritizing treatment based on correctly identifying the severity of a disease or an injury also ensures patients’ safety. The five actual clinical scenarios in the program helped to promptly classify patients into five clinical situations based on the 2021 Korean Emergency Patient Classification Tool. Seventeen nursing students were in an experimental group that had access to a virtual reality-based simulation combined with clinical practice. Seventeen nursing students were in a control group that only participated in routine clinical practice. The virtual reality-based nursing education program effectively improved students’ severity classification competency, performance confidence, and clinical decision-making ability. Although the pandemic continues, the virtual reality-based nursing education program provides realistic indirect experiences to nursing students in situations where clinical nursing practice is not possible. In particular, it will serve as basic data for the expansion and utilization strategy of virtual reality-based nursing education programs to improve nursing capabilities.

## 1. Introduction

Severity classification in emergency care helps efficiently treat and care for emergency room (ER) patients despite limited resources [1,2]. Severity classification was mainly introduced to increase the efficiency of treatment due to overcrowding in the ER and to prevent treatment delays [3]. In the post-2020 COVID-19 pandemic era, we realized that severity classification competency is a major skill necessary not only for ER nurses, but also for general ward nurses, because the severity of many general ward and intensive care unit (ICU) patients may change suddenly and intensify rapidly [4,5]. Most general ward nurses found it difficult to overcome the COVID-19 pandemic because they were not as competent in severity classification [6]. Therefore, it is necessary to develop systematic learning to build severity classification competency, beginning from the undergraduate course of nursing colleges [7,8,9].

Currently, most of the studies that have educated healthcare workers on severity classification and verified its effectiveness have been conducted on emergency medical technicians [9,10]. Additionally, studies that systematically educated nurses on severity classification and verified the education methods’ effectiveness were only partially conducted with military nursing personnel and ER nurses [7,11]. Particularly, most severity classification-related education for nursing students comprises lectures, case studies, case-based small-group learning methods, and simulation education using high-fidelity simulations [12,13,14]. However, the ability to classify the severity of emergency patients cannot be effectively acquired with theoretical knowledge transfer through simple lecture-based learning [7]. Furthermore, owing to COVID-19, clinical practice has been replaced by in-school or online practice, which has made it necessary to introduce changes in education that are suitable for the post-COVID-19 era, such as increasing demands for a safe clinical practice-related environment [15]. To overcome these limitations and apply appropriate educational methods to improve students’ combined effectiveness, interest in practical education using virtual reality (VR), one of the core technologies of the Fourth Industrial Revolution, has been increasing [16].

Severity classification education should be structured to expose students to real-life emergencies and to improve awareness, knowledge, and problem-solving skills related to emergency patient severity classification [7,17]. In the case of high-fidelity simulations currently conducted at nursing universities, it is difficult to implement the clinical situation diversely and practically, and the ratio of instructors to students is low. Further, this learning method cannot provide students with enough opportunities for repetitive training due to space-constraint in the simulation room, so there is a limit to enhancing proficiency in nursing skills [18,19]. Similarly, in suburban clinical practice, nursing students are limited to simple and low-risk nursing activities focused on observation in emergency situations and experience limitations in cultivating their ability to classify severity [19].

Today, information delivery methods are not simply a one-way provision of visual data but are advancing in an interactive way that stimulates the five senses using augmented reality, VR, holograms, and so on, allowing learners and educators to interact [20,21]. Among these, VR utilization in education has the advantage of maximizing the user’s visual experience and immersion with the synesthetic multi-needs of the five senses [22,23,24,25]. Simulation training with the help of VR is a representative realistic media tool that enables users to have a more immersive experience of a given situation. Additionally, VR-based simulation education has the advantage of offering the scope to repeatedly learn nursing interventions, including nursing skills needed to address nursing problems [18,22].

Previous studies showed that VR-based simulation education improves the skill, knowledge, and educational satisfaction of nursing students [26]. This improvement continued even after two months [27]. Additionally, such education positively affects performer confidence, clinical decision-making ability, critical thinking ability, and participation [28,29,30,31]. However, most studies are single-patient, disease-oriented simple-scenario application studies [32,33]. Studies on emergency management VR-based simulation nursing education programs remain insufficient. Training with the help of VR, which is mainly conducted in nursing universities, has increased practical satisfaction; education can be conducted regardless of place and time, but a lack of interaction increases due to the lack of immediate feedback from instructors [29]. Interaction between learners and instructors is one of the ways to enhance educational effectiveness, especially correction feedback that corrects mistakes and has a positive effect on improving self-efficacy and performance confidence [34,35]. Therefore, strategies to enhance students’ immersion in simulation-based education design using VR, as well as providing interaction and feedback on wrong behavior when nursing students classify patient severity are important for improving related knowledge, severity classification competency, and clinical performance.

Therefore, this study verified the effect on nursing students’ knowledge, confidence, and clinical performance ability of a VR-based nursing education program aimed at improving their severity classification competency. This three-dimensional (3D) VR also enhances interaction between motion function and feedback. This program provides realistic indirect experiences to nursing students in situations where clinical nursing practice is not possible. This can be used as a guide when designing and developing nursing education programs using VR.

This study aimed to develop a VR-based nursing education program for improving nursing college students’ ability to classify severity and to verify the program’s effectiveness. By maximizing students’ immersion and liveliness through programs like this, we want to increase their competency to classify severity correctly in the event of multiple emergency patients.

## 2. Materials and Methods

### 2.1. Design

This is a pre- and post-quasi-experimental study with two comparison groups.

### 2.2. Participants

This study was conducted with fourth-grade students who had more than one year of clinical practice experience at K University in D Metropolitan City, Korea. Those who had no problems with vision and hearing, understood the study purpose and method through bulletin boards or department notices in schools, and voluntarily agreed to participate in the study were included as participants. The sample size required for the comparison of two groups (ANOVA) was calculated using the G power version 3.1 program and was based on a study of a nursing education program using a virtual reality intervention [28]. The effect size was calculated as 0.25, power (1-β error probability) 0.80, and significance level α 0.05. As a result, the sample size of each group was considered to be 17. There were 22 participants in the experimental group and 22 in the control group, considering a 20% dropout rate. In order to divide the research subjects into the experimental group and the control group, the random assignment function in the IBM SPSS/WIN 27.0 statistical program was used, and the experimental group was set to 1 and the control group to 0, and 22 subjects were automatically assigned to each group. After filling out the questionnaire and before the experiment, 3 of the 22 experimental groups lost contact, and 2 could not participate due to personal reasons, so the number of experimental groups who finally participated in the study was 17. Of the 22 control subjects, 1 lost contact, and 4 did not participate due to personal reasons, so the total number of control subjects who participated in the study was 17. Thus, there were 34 study participants in total. This study was conducted in a single-blind test in which the subjects did not know whether they were in the experimental group or the control group, and only the researcher was allowed to know (Figure 1).

### 2.3. Measures

In this study, the severity classification competency, confidence in performance, and clinical decision-making ability were measured in a preliminary survey. In the post-investigation phase, the severity classification competency, performer confidence, clinical decision-making ability, class evaluation, simulation design evaluation, and class commitment were measured.

#### 2.3.1. Severity Classification Competency

Severity classification competency was measured using the severity classification competency measurement tool developed by Moon [36]. It has 30 items: clinical judgment (13 items), expert circumstances (4 items), operation (4 items), personal coping (4 items), and communication (5 items). Each item is rated on a five-point Likert scale (1 = “not at all” to 5 = “always”). The higher the score, the higher the severity classification competency. At the time of tool development, the reliability of Cronbach’s α was 0.91, and the reliability of this study was 0.96.

#### 2.3.2. Confidence in Performance

Confidence in performance was measured using the Confidence in Performance in Nursing Scale developed by Kim [37], comprising 10 items, including assessment (4 items), diagnosis (2 items), intervention (3 items), and evaluation (1 item). This instrument uses a five-point Likert scale rating system (1 = “not at all” to 5 = “very likely”). The higher the score, the higher the confidence. Cronbach’s ⍺ was calculated to be 0.80 by Kim [37] and 0.94 in this study.

#### 2.3.3. Clinical Decision-Making Ability

Clinical decision-making ability was measured using the Clinical Decision-Making in Nursing Scale (CDMNS) developed by Jenkins [38] and translated and modified by Baek [39]. It has 40 items categorized into 4 groups: search for alternatives or options (10 items), canvassing of objectives and values (10 items), evaluation and reevaluation of consequences (10 items), and search for information and unbiased assimilation of new information (10 items). Each item is rated on a five-point Likert scale. A higher score indicates a higher level of decision-making ability. In Baek’s [39] study, Cronbach’s ⍺ was calculated to be 0.77 and was 0.92 in this study.

#### 2.3.4. Class Evaluation

Class evaluation was measured using a lecture evaluation instrument for each class type developed by Ko et al. [40] and revised by Yoo [41], according to the simulation. It has 12 items related to class operations (2 items), teaching methods and materials (5 items), objectivity of evaluation (3 items), and class satisfaction (2 items). Items are scored on a five-point Likert scale, with higher scores indicating better class evaluation. Cronbach’s ⍺ was 0.89 in Yoo’s [41] study and 0.93 in this study.

#### 2.3.5. Simulation Design Evaluation

To evaluate the simulation design, this study used the Simulation Design Scale, which was developed by the National League for Nursing and revised by Yoo [41]. It comprises 21 items, including learning goals and education (6 items), support (4 items), problem solving (5 items), feedback (4 items), and realism (2 items). Items are scored on a five-point Likert scale, with higher scores indicating better simulation design. Cronbach’s ⍺ was 0.90 in Yoo’s study [41] and 0.97 in this study.

#### 2.3.6. Practice Flow

Practice flow was measured with the flow measurement instrument developed by Engeser and Rheinberg [42] and revised by Yoo [41]. It has 10 items, namely proficiency in performance (6 items) and immersion in action (4 items). It uses a five-point Likert scale. The higher the score, the more immersive the practice. Cronbach’s ⍺ was 0.84 in Yoo’s study [41] and 0.96 in this study.

### 2.4. Data Collection

Data were collected from 26 April to 30 June 2022, with 17 fourth-year nursing students in the experimental group and 17 students in the control group. In this study, in order to prevent errors in the experimental diffusion effect, the study was conducted first with the control group, with a measurement interval of 1–2 weeks, and then the experimental group was conducted [28]. The control group collected data from 26 April to 20 May, and the experimental group collected data from 7 June to 30 June, two weeks later. Prior to data collection, the participants were informed of the study purpose, method, processes, and their rights in detail.

Before entering the education room, information regarding the participants’ general characteristics, confidence in performance, and clinical decision-making ability were collected using survey questionnaires.

### 2.5. Development of a Nursing Education Program to Improve the Severity Classification Competency of Nursing Students Using Virtual Reality

This study was based on the Analysis-Design-Development-Implementation-Evaluation (ADDIE) model, a general model of teaching method development to develop nursing education programs that promote severity classification competency using VR.

First, in the analysis stage, educational content related to severity classification was designed based on literature analysis and interviews with nurses, learners, and instructors. The literature search was conducted using Web of Science, Cumulative Index of Nursing and Allied Health Literature (CINAHL), PubMed, Research Information Sharing Service (RISS), simulation standards, emergency nursing, and medical textbooks. When using the search engines, the main keywords were used by combining “nursing”, “virtual simulation”, “nursing” students, and “simulation education” tree competency. An interview was conducted to collect basic data on the design of the educational program. Fourth-year students who completed emergency nursing courses and had clinical practice experience, nurses working in the ER, and nursing professors teaching emergency nursing participated. The interview was conducted after explaining the purpose and procedure of the study through wire communication to the participants who met the inclusion criteria. The interview question was, “What should be included in the development of a nursing education program to promote the ability of nursing students to classify severity?”

Second, in the design stage, the learning goals and methods of the nursing education program were designed. Referring to the demand analysis and prior literature, the learning goal was to strengthen the severity classification capability of nursing students. The learning operation method is a nursing education program that uses VR based on the Jeffries [43] model. The program was organized to draw learners to the first step, the participation process, by applying the 5E circular learning model and to enable classification of severity by applying prior knowledge of key concepts. The next step was to learn the knowledge and skills related to severity classification by conducting correct severity classification in emergency situations through exploration, explanation, and expansion processes. The five actual clinical scenarios consist of one of the five levels (Resuscitation, Emergency, Urgency, Less urgency, and Non-urgency) classified based on the 2021 Korean Emergency Patient Classification Tool (KTAS). Each scenario is designed with nursing content that quickly classifies emergency patients to their level. There are five specific contents: Level 1 (Adult Patient-CPR), Level 2 (Adult Patient-Heartache), Level 3 (Adult Patient-Fever), Level 4 (Adult Patient-Laceration), and Level 5 (Adult Patient-Psychiatric Special Situation). The contents of the scenario were produced as 3D images through a 360-degree camera, and based on this, a VR-based nursing education program was developed. Each of the five scenarios consists of a 3 min video and a 2 min quiz and commentary. If all scenarios are answered correctly, the minimum running time of the program is 15 min. If students choose an incorrect answer, a hint is provided, and the quiz is repeated again. The time limit for running the program for each student is 30 min. It was designed to enable interaction within a virtual environment and experiencing it freely anywhere using head-mounted display (HMD) devices.

Additionally, learners can actively use their severity classification knowledge, technology, and strategy through the process of classifying five patient cases in a VR environment and transform and readjust false knowledge into a new form through feedback provided when an incorrect answer is chosen. This VR interaction increased the learners’ sense of immersion and maximized their sense of reality in the actual field. The final stage was designed as a self-evaluation and a reflection diary as an evaluation process. Oculus Quest 2 (Meta Platforms, Inc., CA, USA) was used as the learning medium, and tools for the classification of severity classification competency, confidence in performers, clinical decision-making ability, class evaluation, simulation design evaluation, and practical immersion evaluation were selected.

Third, in the development stage, after developing the learning materials, the content validity was verified by experts consisting of 2 adult-nursing professors and 3 clinical nurses, with more than 10 years of clinical experience in ER nursing. Five fourth-year students from universities that the study participants did not attend were selected. The program was revised and supplemented after the pilot application to develop the final learning materials.

### 2.6. Application of Nursing Education Program to Improve Severity Classification Competency of Nursing Students Using Virtual Reality

#### 2.6.1. Preliminary Survey

As part of a preliminary survey, the two groups were asked to fill in information about general characteristic items, such as age, preferred education method, the previous semester’s grades, severity classification competency, performance confidence, and clinical decision-making ability.

#### 2.6.2. Application of Nursing Education Program to Improve Severity Classification Competency

The experimental group was administered the nursing education program through Oculus Quest 2, which entails an audio-visual format that completely blocks out any surrounding stimuli. For two hours in the first week, a lecture on the theory of severity classification was conducted, and the nursing education program was conducted based on the 5E learning cycle model.

Oculus Quest 2 entails an audio-visual format that completely blocks out any surrounding stimuli. It consists of a design reinforced with motor sensory functionality to enable hands-on activities in VR [16]. HMD equipment reinforced with motor sensory functions provides a mobility interface for position and orientation in addition to visual and auditory interfaces, resulting in natural interactions with virtual environments [16]. This allowed learners to turn their heads and freely look at the surrounding environment, feel visual vividness, and experience auditory vividness through the provided voice. They accessed Oculus Quest 2 individually to familiarize themselves with the learning goals and methods of program driving. Subsequently, severity classification for five clinical cases was conducted. The time limit for running the program for each student is 30 min, and they were notified in advance that the program will be terminated if it exceeds this limit.

The clinical case provided in this study is a severity classification situation that occurs around a one-on-one relationship between a nurse and a patient and was developed to allow nurses to perform severity classification through interaction with a patient. The process of classifying severity was organized to enable learners to directly select the severity classification through a 3D controller. The related concepts or clues were provided using a pop-up window with feedback according to their choice. On selecting the correct answer, one can see an image and hear a voice for the commentary. When a wrong answer is selected, visual data and voice are prompted with the hint, and then the question page can be returned to. In this study, feedback was provided through pop-up windows and auditory resources. The overall feedback was provided during the debriefing after the program was terminated. As the learner was engaging with the education program, the researcher mirrored the screen provided to the learner on a laptop to monitor the learner’s learning process in real time.

The control group participants were presented with the same five clinical scenarios as the experimental group, and they applied them to perform severity classification through case-based learning. This was conducted through face-to-face learning as in general lecture-style education. The instructor acted as an observer without intervening in the learning process, allowing students to solve problems on their own. Feedback was provided during the debriefing time.

#### 2.6.3. Post Program Survey

In the post program survey, the two groups were surveyed in terms of the severity classification competency, performance confidence, clinical decision-making ability, practical immersion, class evaluation, and simulation design evaluation using questionnaires.

### 2.7. Data Analysis

The data collected in this study were analyzed using IBM SPSS/WIN 27.0. The general characteristics of the participants were analyzed using numbers and percentages, mean, and standard deviation. The homogeneity verification of the general characteristics of the participants was analyzed using the x2 test and Fisher’s exact test. The homogeneity verification of performance confidence and clinical decision-making ability was analyzed using an independent *t*-test. The effectiveness of the two groups’ performance confidence and clinical decision-making ability pre-and post-education was analyzed using repeated measures of analysis of variance (ANOVA). The differences between the two groups’ post-education class evaluations, simulation design evaluations, and practical immersion were analyzed using an independent *t*-test. The reliability of the tool was verified using Cronbach’s α.

## 3. Results

### 3.1. General Characteristics and Homogeneity Test of Participants

Table 1 and Table 2 shows the participants’ general characteristics and homogeneity verification. This study included 34 participants, with 17 in the experimental group and 17 in the control group. The average age was 21.65 years in the experimental group and 21.59 years in the control group. As for participants’ preferred education methods, the largest number of students answered that they preferred lecture education, with 11(64.7%) in the experimental group and 11 students (64.7%) in the control group. For the prior semester’s grades, the largest number of students answered that they received a grade of 3.5 or higher but less than 4.0, with 13 (76.4%) in the experimental group and 11 (64.7%) in the control group. The results of homogeneity for general characteristics and dependent variables showed no significant difference between the two groups in terms of age, preference for education, and previous semester grades, severity classification capability, or performance confidence, indicating that the two groups were homogeneous (*p* > 0.05). However, the clinical decision-making ability was homogeneous only in terms of “Evaluation and reevaluation of consequences” in the case of the sub-disciplinary subjects (*t* = −0.30, *p* = 0.760) (Table 2).

### 3.2. Difference between the Experimental Group and the Control Group’s Severity Classification Competency, Performance Confidence, and Clinical Decision-Making Ability before and after the Program

Table 3 shows the results of verifying the difference between the experimental and control groups’ severity classification competency, performance confidence, and clinical decision-making ability before and after the program. The severity classification competency of the experimental group was 2.71 (±0.53) points before the program and 4.12 (±0.26) after the program, higher than the control’s average of 2.95 (±0.364) and 3.88 (±0.48) post program, which was statistically significant (F = 5.73, *p* = 0.023). The experimental group averaged 3.43 (±0.46) before the program and 4.18 (±0.29) post program, higher than the control group’s 3.81 (±0.39) and 4.06 (±0.56) after the program, indicating statistically significant differences between performance confidence and group-time interactions (F = 8.63 *p* = 0.006). The canvassing of objectives and values of clinical decision-making ability of the experimental group was 3.95 (±0.40) after the program and 3.28 (±0.25) before the program. The control group’s clinical decision-making ability was 3.57 (±0.45) post program. The difference between the two groups was statistically significant (F = 7.17, *p* < 0.001).

### 3.3. Differences in Class Evaluation, Simulation Design Evaluation, and Practice Immersion between the Groups Post Program

Table 4 shows the results of the comparison of the post-program class evaluation, simulation design evaluation, and Practice flow between the experimental and control groups. The average post-program class evaluation of the experimental group was 4.78 (±0.29), lower than the average of 4.82 (±0.39) of the control group, but it was not statistically significant (t = −0.333, *p* = 0.741). The post-program simulation design evaluation in the experimental group averaged 4.67 (±0.32), which was lower than the average of 4.69 (±0.55) of the control group, but it was not statistically significant (t = −0.073 *p* = 0.942). The practice flow in the experimental group had an average of 4.31 (±0.48), which was lower than the average of 4.46 (± 0.66) of the control group, but it was not statistically significant (t = −0.773, *p* = 0.456).

## 4. Discussion

This study developed a VR-based nursing education program for nursing college students to improve their severity classification and verified its effectiveness. The results showed that this program effectively improved performance confidence, severity classification competency, and clinical decision-making ability.

This program increased the sense of realism and immersion by filming and producing VR videos based on emergency cases using a 3D camera. Information regarding severity classification was provided, with several cases corresponding to stage 5 according to the KTAS criteria [7,8]. It enabled repeated learning of various patient situations. The captured video was produced by applying editing and motion interaction design using a haptic device. In VR utilization studies [44,45,46], it has been reported that the use of VR in nursing education provides a realistic virtual environment, increasing immersion and enhancing interaction through actions using haptic devices, which are efficient in maximizing technical training and learning effects [16]. Therefore, the provision of clinical field-based VR and the design of a nursing education program for interaction and repetitive learning in this study are believed to have helped maximize students’ learning focused on classifying the severity of clinical patients. Furthermore, the learning process in this study used an HMD based on the 5E learning process to maximize engagement in learning by providing a lively situation against the background of a 3D clinical site. Additionally, the participating students explained a given clinical situation and went through the elaborate step of determining the severity of the patient using their previous knowledge. Additionally, the program includes an Elaborate phase in which participating students explore a given clinical situation and use their prior knowledge to determine the patient’s severity. At the final evaluation phase, participating students are provided with feedback on their performance. Through this process, learners will be able to correct their wrong knowledge and nursing skills. The stepwise adaptation of the 5E cyclical model of the VR program suited the emphasis on allowing students to understand through the presented series of established steps or stages [28,47,48].

There were no statistically different variables compared to case-based learning in the design evaluation score and the learning satisfaction score of the nursing education program, but both evaluation scores were higher than 4.5 out of 5. Prior studies using VR [49,50] indicated low learning satisfaction because they could not provide images using simple animations to demonstrate the vitality of the clinical field and design based on systemized learning strategies, such as learning models [51,52]. The step-by-step design centered on the 5E learning cycle model of this nursing education program enhanced students’ immersion and helped them acquire knowledge and skills [47,53]. Hence, it yielded higher scores in learning satisfaction and program design evaluation than other studies [49]. Additionally, a study by Yun and Choi [53] showed that case-based learning design increases learning satisfaction by experiencing the nursing situations that cannot directly be experienced in clinical situations, supporting the high learning satisfaction of VR-based nursing education programs. Therefore, it is necessary to consider developing various learning programs that can provide indirect clinical experience based on clinical scenarios to enhance the clinical nursing capacity of nursing students at a time when their clinical environment has decreased due to changes in the clinical environment, such as patient rights issues and infectious diseases.

The VR-based nursing education program to promote severity classification competency was effective in enhancing confidence performance, severity classification competency, and clinical decision-making ability compared with the case-based learning method. In a meta-study using VR in nursing education, it was reported that the use of VR in nursing education does not actually exist but provides a sense of immersion in a 3D virtual world, which is effective in shortening the training time and improving the knowledge of nursing students [15,54]. Particularly, VR-based education links previously learned concepts through interactive learning when compared to traditional education, which helps to integrate new knowledge and apply it to real-world situations [55]. In a review study [56] that analyzed studies using VR to improve nursing skills, it was reported that “virtual reality education was not more effective in providing skills than other simulation learning”. This suggests that high-fidelity simulation education using mannequins rather than simulation education using VR in non-face-to-face education is more efficient in acquiring nursing skills, by providing students with an opportunity to practice performance directly. However, in this study, the classification of patient severity was not a technique in which complex nursing procedures were learned and performed, but a clinical performance ability to analyze patient situations and give them canvassing of objectives and values. Therefore, in this study, the clinical performance ability related to not only knowledge, but also to the canvassing of objectives, and value judgment was improved through a nursing education program using VR. This is believed to have been effective in improving severity classification competency and canvassing of objectives and value ability by integrating knowledge using VR and applying interactive motion technology, rather than case-based learning methods. However, to improve the severity classification competency of nursing students, it will be necessary to compare the effects of high-fidelity simulation using mannequins and evidence competency in the future. In this study, only the “canvassing of objectives and value” item among the sub-areas of clinical decision-making ability was effective compared to case-based education. The canvassing of objectives and value of clinical decision making induces nurses to find and solve patients’ health problems with open and positive values in the process of selecting and applying challenging alternatives based on clinical experience [57]. Therefore, among the clinical decision-making capabilities regarding severity classification, canvassing of objectives and value is an essential competency to judge and solve a patient’s health problems.

The limitations of this study are as follows. First, the generalizability of the results is limited due to the study population of nursing students at a nursing university. Second, there is a limitation in evaluating the persistence of the effect of the VR-based nursing education program because re-examination after intervention was not conducted as a measurement parameter before and after intervention. Lastly, in this study, most of the measurement of effectiveness of the education program in improving severity classification competency relied on self-report questionnaires, and it is difficult to objectively evaluate the effectiveness of the nursing education program, because only five actual clinical scenarios was applied.

As a result of the above research, a VR program to learn the severity classification of patients was developed to learn the emergency patient classification tool, so it can be effectively used in the field. In addition, if the contents of learning are continuously updated by reflecting changes in the clinical field, the learning effect of emergency patient classification can be further improved. Based on the results of this study, we would like to suggest the following. A study that can confirm the VR program’s continuous effect on learning the severity classification of patients and a repeated study to verify the learning effect by developing additional scenarios and learning videos for various severity classification situations are suggested.

## 5. Conclusions

The designed program in this study helped improve the values of severity classification competency and clinical decision-making ability by integrating previously learned knowledge. Accordingly, it is believed that the use of VR in nursing education will provide a realistic clinical field to integrate previously learned concepts and help develop clinical judgment skills. This study’s educational design was developed by applying VR to learn the severity classification of patients. However, if VR is used as a nursing education strategy in various clinical situations that require clinical judgment in the future, it can be used to integrate the educational content into knowledge and make accurate clinical judgments.

## Figures and Tables

**Figure 1 healthcare-11-01122-f001:**
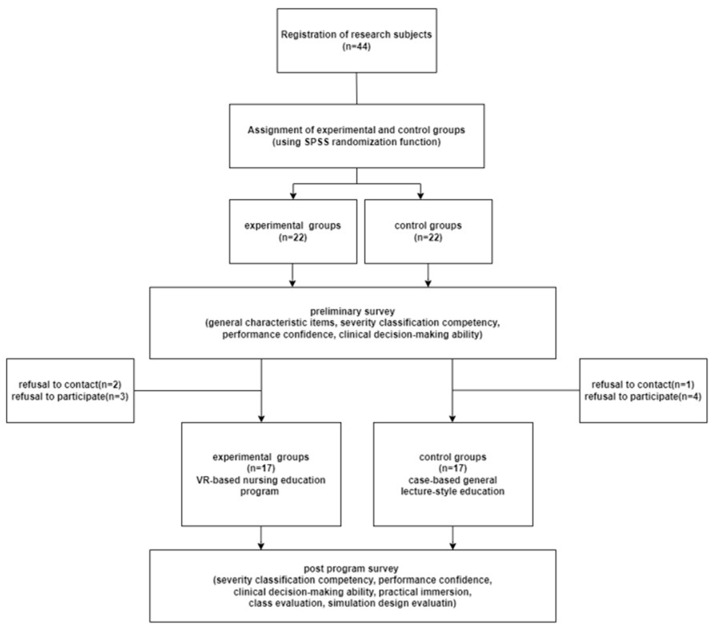
Flow chart of the study.

**Table 1 healthcare-11-01122-t001:** General Characteristics of the Participants (*n* = 34).

Characteristics	Categories	Exp.(*n* = 17)	Cont.(*n* = 17)	x^2^ or F	*p*
		*n* (%) or M ± SD	*n* (%) or M ± SD		
Age (year)		21.65 ± 0.61	21.59 ± 0.87	3.04	0.091
Preferred education	Lecture	11(64.7)	11(64.7)	0.58	0.452
	Discussion	1(5.9)	1(5.9)		
	Q and A	2(11.8)	0(0)		
	Performance	3(17.6)	5(29.4)		
Last semester grade	<2.5	1(5.9)	1(5.9)	2.04	0.163
	2.5–3.0	2(11.8)	1(5.9)		
	3.0–3.5	10(58.8)	5(29.4)		
	3.5–4.0	3(17.6)	6(35.3)		
	4.0–4.5	1(5.9)	4(23.5)		

**Table 2 healthcare-11-01122-t002:** Pretest Homogeneity between the Experimental and Control Groups (*n* = 34).

Variables	Exp.(*n* = 17)	Cont.(*n* = 17)	*t*	*p*
	M ± SD	M ± SD		
Severity classification competency	2.71 ± 0.53	2.95 ± 0.39	1.45	0.157
Confidence in performance	3.43 ± 0.46	3.81 ± 0.39	−0.80	0.427
Clinical decision-making ability	3.28 ± 0.18	3.66 ± 0.66	4.59	<0.001
Search for alternatives or options	3.27 ± 0.28	3.77 ± 0.51	3.59	<0.001
Evaluation and reevaluation of consequences	3.44 ± 0.31	4.05 ± 0.38	5.09	<0.001
Canvassing of objectives and values	3.28 ± 0.41	3.25 ± 0.41	−0.30	0.760
Search for information and unbiased assimilation of new information	3.15 ± 0.29	3.55 ± 0.43	3.17	<0.001

**Table 3 healthcare-11-01122-t003:** Pretest–Posttest Mean Differences of Confidence in Performance and Clinical Decision-Making Ability between the Experimental and Control Groups Over Time (*n* = 34).

Characteristics	Exp.(*n* = 17)	Cont.(*n* = 17)	Source	x^2^ or F	*p*
	M ± SD	M ± SD			
**Confidence in performance**					
Pretest	3.43 ± 0.46	3.81 ± 0.39	Group	1.07	0.309
Posttest	4.18 ± 0.29	4.06 ± 0.56	Time	34.92	<0.001
			Group/time	8.631	0.006
**Severity classification competency**					
Pretest	2.71 ± 0.53	2.95 ± 0.36	Group	0.01	0.957
Posttest	4.12 ± 0.26	3.88 ± 0.48	Time	134.99	<0.001
			Group/time	5.73	0.023
**Clinical decision-making ability_Canvassing of objectives and values**					
Pretest	3.28 ± 0.25	3.25 ± 0.40	Group	3.31	<0.001
Posttest	3.95 ± 0.40	3.57 ± 0.45	Time	56.27	<0.001
			Group/time	7.17	<0.001

**Table 4 healthcare-11-01122-t004:** Mean Differences of Class Evaluation, Simulation Design Evaluation and Practice Flow between the Experimental and Control Groups (*n* = 34).

Variables	Exp.(*n* = 17)	Cont.(*n* = 17)	*t*	*p*
	M ± SD	M ± SD		
**Class evaluation**				
Learning management	4.79 ± 0.36	4.82 ± 0.39	−0.29	0.821
Instructional methods, materials	4.85 ± 0.24	4.84 ± 0.37	0.11	0.913
Objectivity of assessment	4.57 ± 0.65	4.78 ± 0.44	−1.12	0.269
Learning satisfaction	4.91 ± 0.26	4.82 ± 0.39	0.76	0.449
Total	4.78 ± 0.29	4.82 ± 0.39	−0.33	0.741
**Simulation design evaluation**				
Learning goals and education	4.56 ± 0.47	4.55 ± 0.68	0.049	0.961
Support	4.87 ± 0.22	4.71 ± 0.56	1.10	0.280
Problem solving	4.72 ± 0.28	4.73 ± 0.52	−0.81	0.936
Feedback	4.56 ± 0.55	4.79 ± 0.52	−1.27	0.210
Realism	4.74 ± 0.53	4.71 ± 0.59	0.153	0.880
Total	4.67 ± 0.32	4.69 ± 0.55	−0.07	0.942
**Practice flow**	4.31 ± 0.48	4.46 ± 0.66	−0.77	0.456

## Data Availability

Not available.

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
