# Peer review of "Effectiveness of the Patient’s Severity Classification Competency Promotion Virtual Reality Program of Nursing Students during the COVID-19 Pandemic Period"

_healthcare, 2023, doi:10.3390/healthcare11081122_

Round 1
Reviewer 1 Report
In general, this manuscript is well written and is clear. Please see the comments below:
Introduction-The literature is current, and it is presented logically.
Method section:
Could you provide a detailed description of the person who eliminated?
Could you tell me more about the nursing education program?
What is the total time required to learn 5 scenarios?
Are you saying that the VR-based nursing education program takes 15 minutes per scenario?
What is the number of hours per student in the experimental group?
Could you tell me when the 17 students in the experimental group started and ended the experiment?
Did you train the 17 control group students face-to-face after the experimental group training ended?
Did the control group students also receive 1:1 training?
Measures-It is clear.
Data analysis-It is clear.
Results- Could you check this sentence (The post-program simulation design evaluation in the experimental group averaged 4.67 ( 0.32), which was lower than the average of 4.63 (0.55), but was not statistically significant....) This is an incorrect interpretation (line 366).
Discussion - It is clear.
Limitation -Please be specific about the limitations of your study. Did the control group not receive the VR-based nursing education program that the experimental group received?
Conclusion - It is clear.
Author Response
Response to Reviewer 1 Comments
Point 1:
In general, this manuscript is well written and is clear. Please see the comments below:
Introduction-The literature is current, and it is presented logically.
Response 1:
Thank you
Point 2:
Method section:
Could you provide a detailed description of the person who eliminated?
Response 2:
A new detailed description of the assignment of control groups to experimental groups has been added.
The sample size required for the comparison of two groups (ANOVA)was calculated using the G power version 3.1 program was based on the nursing education program using virtual reality intervention study[28], the effect size was calculated as 0.25, power (1-β error probability) 0.80, and significance level α 0.05. As a result, the sample size of each group was considered to be 17. There were 22 participants in the experimental group and 22 in the control group, considering the 20% dropout rate. In order to divide the research subjects into the experimental group and the control group, the random assignment function in the IBM SPSS/WIN 27.0 statistical program was used, and the experimental group was set to 1 and the control group to 0, and 22 subjects were automatically assigned to each group. After filling out the questionnaire and before the experiment, 3 of the 22 experimental groups lost contact and 2 could not participate due to personal reasons, so the number of experimental groups who finally participated in the study was 17. Of the 22 control sub-jects, 1 lost contact and 4 did not participate due to personal reasons, so the total number of control subjects who participated in the study was 17. Thus, there were 34 study par-ticipants in total. This study was conducted in a single-blind test in which the subjects did not know whether they were in the experimental group or the control group, and only the researcher was allowed to know(Figure 1).
Point 3:
Could you tell me more about the nursing education program?
What is the total time required to learn 5 scenarios?
Are you saying that the VR-based nursing education program takes 15 minutes per scenario?
What is the number of hours per student in the experimental group?
Response 3:
A detailed description of the nursing education program has been added.
->Each of the five scenarios consists of a 3-minute video and a 2-minute quiz and commentary. If all scenarios are answered correctly, the minimum running time of the program is 15 minutes. If you choose an incorrect answer, a hint is provided and the quiz is repeated again. The time limit for running the program for each student is 30 minutes.
->Subsequently, severity classification for five clinical cases was conducted. The time limit for running the program for each student is 30 minutes, and it was notified in advance that the program will be terminated if it exceeds this limit
Point 4:
Could you tell me when the 17 students in the experimental group started and ended the experiment?
Response 4:
We have added the relevant part to the data collection in detail.
->In this study, in order to prevent errors in the experimental diffusion effect, the study was conducted first with the control group, with a measurement interval of 1-2 weeks, and then the experimental group was conducted[28]. The control group collected data from April 26 to May 20, and the experimental group collected data from June 7 to June 30, two weeks later.
Point 5:
Did you train the 17 control group students face-to-face after the experimental group training ended?
Did the control group students also receive 1:1 training?
Response 5:
The control group received traditional training to triage cases.This is an instructor-led training, not a 1:1 training.
->The control group participants were presented with the same five clinical scenarios as the experimental group, and they applied them to perform severity classification through case-based learning. This was conducted through face-to-face learning as a general lecture-style education.
Point 6:
Measures-It is clear.
Data analysis-It is clear.
Response 6:
Thank you
Point 7:
esults- Could you check this sentence (The post-program simulation design evaluation in the experimental group averaged 4.67 ( 0.32), which was lower than the average of 4.63 (0.55), but was not statistically significant....) This is an incorrect interpretation (line 366).
Response 7:
Sorry. We re-corrected the missing words.
->The post-program simulation design evaluation in the experimental group averaged 4.67 (±0.32), which was lower than the average of 4.69 (±0.55) of the control group, but was not statistically significant (t = -.073 p = .942). The practive flow in th experimental group had an average of 4.31 (±0.48), which was lower than the average of 4.46 (±0.66) of the control group, but it was not statistically significant (t = -.773, p = .456).
Point 8:
Discussion - It is clear.
Response 8:
Thank you
Point 9:
Limitation -Please be specific about the limitations of your study. Did the control group not receive the VR-based nursing education program that the experimental group received?
Response 9:
We added limitations of the study.
->The limitations of this study are as follows. First, the generalizability of the results is limited due to the study population of nursing students at a nursing university. Second, there is a limitation in evaluating the persistence of the effect of the VR-based nursing education program because re-examination after intervention was not conducted as a measurement parameter before and after intervention. Lastly, most of the severity classification competency by the effectiveness of the education program in this study relied on self-report questionnaires, and it is difficult to objectively evaluate the effectiveness of the nursing education program because only five actual clinical scenarios was applied.
Point 10:
Conclusion - It is clear.
Response 10:
Thank you

Reviewer 2 Report
First of all, I want to congratulate the authors of this paper. The topic discussed is very relevant and a study that tests the effectiveness of new teaching methodologies in undergraduate nursing is of great interest to readers.
The introduction focuses the research topic very well.
The methodology is pertinent and correctly detailed in the manuscript.
I find the sample to be small; I recommend including a section on the limitations of the study to reflect this aspect.
The results are presented in a clear manner, although I suggest that the authors reduce the explanation of each table as it is repetitive.
The discussion is well constructed and the conclusions are consistent with the results obtained.
A very timely and necessary investigation. Congratulations.
Author Response
Response to Reviewer 2 Comments
Point 1:
First of all, I want to congratulate the authors of this paper. The topic discussed is very relevant and a study that tests the effectiveness of new teaching methodologies in undergraduate nursing is of great interest to readers.
The introduction focuses the research topic very well.
The methodology is pertinent and correctly detailed in the manuscript.
Response 1:
Thank you
Point 2:
I find the sample to be small; I recommend including a section on the limitations of the study to reflect this aspect.
Response 2:
We added limitations of the study.
->The limitations of this study are as follows. First, the generalizability of the results is limited due to the study population of nursing students at a nursing university. Second, there is a limitation in evaluating the persistence of the effect of the VR-based nursing education program because re-examination after intervention was not conducted as a measurement parameter before and after intervention. Lastly, most of the severity classification competency by the effectiveness of the education program in this study relied on self-report questionnaires, and it is difficult to objectively evaluate the effectiveness of the nursing education program because only five actual clinical scenarios was applied
Point 3:
The results are presented in a clear manner, although I suggest that the authors reduce the explanation of each table as it is repetitive.
Response 3:
We have briefly modified that part. In addition to the parts below, we have deleted the parts described by overlapping the results part.
The average age was 21.65 years in the experimental group and 21.59 years in the control group. Participants' preferred education methods ; the largest number of students answered that they preferred lecture education with 11(64.7%) in the experimental group and there were 11 students (64.7%) in the control group . In last semester's grades; the largest number of students answered that they received grade of 3.5 or higher and less than 4.0, with 13 (76.4%) in the experimental group and 11 (64.7%) in the control group, respectively. The results of ho-mogeneity for general characteristics showed no significant difference between the two groups in terms of age, preference for education, and previous semester grades, indicating that the two groups were homogeneous (p > .05). As a result of the homogeneity test of the dependent variables of the two groups before implementing the program, the severity classification capability (t = 1.45, p = .427) and performance confidence (t = -.80, p = .427) showed no significant difference, indicating that the two groups were homogeneous. However, the clinical decision-making ability was homogeneous only in terms of “Evaluation and reevaluation of consequences” in case of the sub-disciplinary subjects (t = -0.30, p = .760) (Table2).
Point 4:
The discussion is well constructed and the conclusions are consistent with the results obtained.
A very timely and necessary investigation. Congratulations.
Response 4:
Thank you

Reviewer 3 Report
This is an interesting experimental study using VR for practical education of nursing students. A variety of variables were used and reported in the study results.
1. The background and necessity of the study are convincingly described in the introduction, but it is too long. A shorter description is necessary.
2. What is the design of this study? Is it a quasi-experimental design, a non-equivalent control group pre-test/post-test design?
3. What is the analysis method when calculating with G power version 3.1 program? What is the basis for setting the power to 0.8 and setting the effect size to medium?
4. When the number of samples calculated is 17, 22 are needed if calculated at a dropout rate of 20%. Why did 21 come out?
5. Clinical decision-making ability and class evaluation consisted of 4 sub-items. How many questions did each sub-item consist of?
6. It was said that the two groups were randomly assigned. Please present the CONSORT flow diagram.
7. Please describe how the randomization method was conducted.
8. It was not known which group the subject belonged to, but which method was used as a blind method for the subject? In the case of students from the same school, how did you prevent contamination?
9. In Table 2, the sub-item of Clinical decision making ability is different from the term described in the tooltip, and is not shown as a sub-item in the table. Please modify the table to show more subcategories.
10. Do not duplicate the results in the table, but write more concisely.
11. Discussion AND Conclusion: GOOD
Author Response
Response to Reviewer 3 Comments
Point 1:
- The background and necessity of the study are convincingly described in the introduction, but it is too long. A shorter description is necessary.
Response 1:
Sentences in the introduction have been edited for brevity.
Word count in original introduction: 948
Changed introduction word count: 857
Point 2:
- What is the design of this study? Is it a quasi-experimental design, a non-equivalent control group pre-test/post-test design?
Response 2:
This is a pre- and post-quasi-experimental study with two comparison groups.
Point 3:
- What is the analysis method when calculating with G power version 3.1 program? What is the basis for setting the power to 0.8 and setting the effect size to medium?
Response 3:
A new detailed description of the assignment of control groups to experimental groups has been added.
-> The sample size required for the comparison of two groups (ANOVA)was calculated using the G power version 3.1 program was based on the nursing education program using virtual reality intervention study[28], the effect size was calculated as 0.25, power (1-β error probability) 0.80, and significance level α 0.05. As a result, the sample size of each group was considered to be 17.
Point 4:
- When the number of samples calculated is 17, 22 are needed if calculated at a dropout rate of 20%. Why did 21 come out?
Response 4:
Sorry, I corrected the wrong part listed as 21 people.
-> The sample size required for the comparison of two groups (ANOVA)was calculated using the G power version 3.1 program was based on the nursing education program using virtual reality intervention study[28], the effect size was calculated as 0.25, power (1-β error probability) 0.80, and significance level α 0.05. As a result, the sample size of each group was considered to be 17. There were 22 participants in the experimental group and 22 in the control group, considering the 20% dropout rate.
Point 5:
- Clinical decision-making ability and class evaluation consisted of 4 sub-items. How many questions did each sub-item consist of?
Response 5:
Clinical decision-making ability was measured using the Clinical Decision-Making in Nursing Scale (CDMNS) developed by Jenkins [38] and translated and modified by Baek [39]. It has 40 items categorized into 4 groups: search for alternatives or options (10 items), canvassing of objectives and values (10 items), evaluation and reevaluation of consequences (10 items), and search for information and unbiased assimilation of new information (10 items). Each item is rated on a five-point Likert scale. A higher score indicates a higher level of decision-making ability. In Baek’s [39] study, Cronbach's ⍺ was calculated to be .77 and was .92 in this study.
Point 6:
- It was said that the two groups were randomly assigned. Please present the CONSORT flow diagram.
Response 6:
The relevant information was inserted into the flow chart of the study. Please refer to Figure 1.
Point 7:
- Please describe how the randomization method was conducted.
Response 7:
There were 22 participants in the experimental group and 22 in the control group, considering the 20% dropout rate. In order to divide the research subjects into the experimental group and the control group, the random assignment function in the IBM SPSS/WIN 27.0 statistical program was used, and the experimental group was set to 1 and the control group to 0, and 22 subjects were automatically assigned to each group.
Point 8:
- It was not known which group the subject belonged to, but which method was used as a blind method for the subject? In the case of students from the same school, how did you prevent contamination?
Response 8:
This study was conducted in a single-blind test in which the subjects did not know whether they were in the experimental group or the control group, and only the researcher was allowed to know
In this study, in order to prevent errors in the experimental diffusion effect, the study was conducted first with the control group, with a measurement interval of 1-2 weeks, and then the experimental group was conducted[28]. The control group collected data from April 26 to May 20, and the experimental group collected data from June 7 to June 30, two weeks later
Point 9:
- In Table 2, the sub-item of Clinical decision making ability is different from the term described in the tooltip, and is not shown as a sub-item in the table. Please modify the table to show more subcategories.
Response 9:
Fixed the subcategory to be the same as the research tool description.
Point 10:
- Do not duplicate the results in the table, but write more concisely.
Response 10:
The results have been edited concisely and unnecessary information has been deleted. In addition to the parts below, we have deleted the parts described by overlapping the results part.
The average age was 21.65 years in the experimental group and 21.59 years in the control group. Participants' preferred education methods ; the largest number of students answered that they preferred lecture education with 11(64.7%) in the experimental group and there were 11 students (64.7%) in the control group . In last semester's grades; the largest number of students answered that they received grade of 3.5 or higher and less than 4.0, with 13 (76.4%) in the experimental group and 11 (64.7%) in the control group, respectively. The results of ho-mogeneity for general characteristics showed no significant difference between the two groups in terms of age, preference for education, and previous semester grades, indicating that the two groups were homogeneous (p > .05). As a result of the homogeneity test of the dependent variables of the two groups before implementing the program, the severity classification capability (t = 1.45, p = .427) and performance confidence (t = -.80, p = .427) showed no significant difference, indicating that the two groups were homogeneous. However, the clinical decision-making ability was homogeneous only in terms of “Evaluation and reevaluation of consequences” in case of the sub-disciplinary subjects (t = -0.30, p = .760) (Table2)
Point 11:
- Discussion AND Conclusion: GOOD
Response 11:
Thank you

Round 2
Reviewer 1 Report
I confirmed that the modification worked. I think it is safe to release as is.
Author Response
Response to Reviewer 1 Comments
Point 1:
I confirmed that the modification worked. I think it is safe to release as is.
Response 1:
thank you
Reviewer 3 Report
Thank you for your efforts.
line 245
If you choose an incorrect answer --> It seems that the subject should be students, not 'you'.
line 340~346
Since the results are presented in Table 2, describing only those not homogenous seems necessary.
Please do not duplicate the results in the table, but write more concisely.
'Do not duplicate the results in the table, but write more concisely.'
I think you misunderstood this comment. It means deleting the overlapping part with the table from the result description part.
Author Response
Response to Reviewer Comments
Point 1:
line 245
If you choose an incorrect answer --> It seems that the subject should be students, not 'you'.
Response 1:
We have corrected that part.
->If students choose an incorrect answer, a hint is provided and the quiz is repeated again.
Point 2:
line 340~346
Since the results are presented in Table 2, describing only those not homogenous seems necessary.
Please do not duplicate the results in the table, but write more concisely.
'Do not duplicate the results in the table, but write more concisely.'
I think you misunderstood this comment. It means deleting the overlapping part with the table from the result description part.
Response 2:
We modified the sentence to be more concise as below
->The results of homogeneity for general characteristics and dependent variables showed no significant difference between the two groups in terms of age, preference for education, and previous semester grades, severity classification capability, performance confidence indicating that the two groups were homogeneous (p > .05). However, the clinical decision-making ability was homogeneous only in terms of “Evaluation and reevaluation of consequences” in case of the sub-disciplinary subjects (t = -0.30, p = .760) (Table2).